# Fate of terrigenous organic matter across the Laptev Sea from the mouth of the Lena River to the deep sea of the Arctic interior

Lisa Bröder [a,b]*, Tommaso Tesi [a,b,c], Joan A. Salvadó [a,b], Igor P. Semiletov [d,e,f], Oleg V. Dudarev [e,f], Örjan Gustafsson [a,b]

[a] *Department of Environmental Science and Analytical Chemistry, Stockholm University, Stockholm, Sweden*

[b] *Bolin Centre for Climate Research, Stockholm University, Stockholm, Sweden*

[c] *Institute of Marine Sciences – National Research Council, Bologna, Italy*

[d] *International Arctic Research Center, University Alaska Fairbanks, Fairbanks, USA*

[e] *Pacific Oceanological Institute, Russian Academy of Sciences, Vladivostok, Russia*

[f] *Tomsk National Research Politechnical University, Tomsk, Russia*

*corresponding author: lisa.broder@aces.su.se

**Abstract**
Ongoing global warming in high latitudes may cause an increasing supply of permafrost-
derived organic carbon through both river discharge and coastal erosion to the Arctic
shelves. Mobilized permafrost carbon can be either buried in sediments, transported to the
deep sea or degraded to $CO_2$ and outgassed, potentially constituting a positive feedback to
climate change.
This study aims to assess the fate of terrestrial organic carbon (TerrOC) in the Arctic marine
environment by exploring how it changes in concentration, composition and degradation
status across the wide Laptev Sea shelf. We analyzed a suite of terrestrial biomarkers as
well as source-diagnostic bulk carbon isotopes ($\delta^{13}C$, $\Delta^{14}C$) in surface sediments from a
Laptev Sea transect spanning more than 800 km from the Lena River mouth (< 10 m water
depth) across the shelf to the slope and rise (2000-3000 m water depth). These data provide
a broad view on different TerrOC pools and their behavior during cross-shelf transport. The
concentrations of lignin phenols, cutin acids and high-molecular weight (HMW) wax lipids
(tracers of vascular plants) decrease by 89-99 % along the transect. Molecular-based
degradation proxies for TerrOC (e.g., the carbon preference index of HMW lipids, the HMW
acids/alkanes ratio and the acid/aldehyde ratio of lignin phenols) display a trend to more
degraded TerrOC with increasing distance from the coast. We infer that the degree of
degradation of permafrost-derived TerrOC is a function of the time spent under oxic
conditions during protracted cross-shelf transport. Future work should therefore seek to
constrain cross-shelf transport times in order to compute a TerrOC degradation rate and
thereby help to quantify potential carbon-climate feedbacks.

## 1 Introduction

Amplified global warming in high latitudes has raised growing concern about potential positive carbon-climate feedbacks. Arctic soils store about half of the global soil organic carbon (Tarnocai et al., 2009), with 60 % of this in perennially frozen grounds (Hugelius et al., 2014). With ongoing climate change these vast carbon reservoirs become increasingly vulnerable. Mobilization and transport of old terrigenous organic carbon (TerrOC) into the Arctic Ocean is expected to intensify (Gustafsson et al., 2011) through enhancing river discharge (McClelland et al., 2008) with augmenting sediment loads (Gordeev, 2006; Syvitski, 2002) and accelerating coastal erosion (Günther et al., 2013). This material can be buried in the sediments of the Arctic shelves, transported across the margin towards deeper basins or degraded and re-introduced into the modern carbon cycle as $CO_2$, thereby not only providing a potential positive feedback to global warming, but also causing severe ocean acidification (Semiletov et al., 2016). The fate of permafrost-released TerrOC in the marine environment is thus crucial for future climate projections, yet insufficiently understood (Vonk and Gustafsson, 2013).

The East Siberian Arctic Shelf (ESAS) is with a width of > 800 km the world's largest continental shelf. It receives TerrOC both from the erosion of the East Siberian shoreline, largely consisting of organic-rich, late-Pleistocene ice-complex deposits (Yedoma), and via the Great Russian Arctic Rivers, which drain extensive areas of continuous and discontinuous permafrost. The Laptev Sea is a representative for the TerrOC dominated Siberian shelf seas, since its main organic carbon input originates from substantial coastal erosion (as observed in the Buor-Khaya Bay, Sánchez-García et al., 2011; Semiletov et al., 2011; Vonk et al., 2012) and the Lena River, the main fluvial sediment source for the entire ESAS (Holmes et al., 2002).

Previous studies have focused on near-shore areas and the inner shelf (e.g. Bröder et al., 2016; Charkin et al., 2011; Feng et al., 2015; Karlsson et al., 2011; Salvadó et al., 2015; Sánchez-García et al., 2011; Semiletov et al., 2005, 2012, 2013; Tesi et al., 2014; Vonk et al., 2010, 2012, 2014; Winterfeld et al., 2015b, 2015a). They reported large fractions of old

TerrOC in particulate organic carbon (POC) and surface sediments close to the coast, using
different approaches such as applying carbon-isotope-based source apportionment (e.g.
Gustafsson et al., 2011; Semiletov et al., 2005; Vonk et al., 2010, 2012, 2014; Salvadó et al.,
2015, for the iron-associated OC fraction in the sediment) and by analyzing terrigenous
biomarkers in both surface sediments (e.g. Feng et al., 2013; Stein and Macdonald, 2004;
Tesi et al., 2014, 2016) and POC in the water column (e.g. Charkin et al., 2011; Karlsson et
al., 2011; Winterfeld et al., 2015a). This is the first study that encompasses sampling stations
along the entire transect from the Lena River mouth, across the wide Laptev Sea shelf, to the
continental slope and rise. Our major objective was to gain new insights regarding the
behavior of different TerrOC pools, in particular investigating potential degradation of
permafrost-released material along the land-shelf-basin continuum. The Laptev Sea and
adjacent East Siberian Sea are among the widest continental margins on Earth (Jakobsson
et al., 2004). The resulting long cross-shelf transport and thereby time spent in oxic
sediments might exert first order control on TerrOC degradation (e.g. Keil et al., 2004). Our
study area is thus well suited to test hypotheses on the fate of permafrost carbon in terms of
carbon-climate feedback. We have therefore characterized TerrOC in surface sediments
along the Laptev Sea transect on both bulk and molecular level, exploiting source-diagnostic
bulk carbon isotopes ($\delta^{13}$C, $\Delta^{14}$C) as well as an extensive biomarker suite (lignin phenols and
cutin acids obtained by alkaline CuO oxidation and high-molecular-weight solvent-extractable
lipids, such as *n*-alkanes and *n*-alkanoic acids).

## 2   Material and Methods

2.1 Study area

The Laptev Sea is the shallowest of the Arctic shelf seas with an average depth of 48 m

(Jakobsson et al., 2004). It spans over 498,000 km$^2$ with a volume of 24,000 km$^3$ and is

located between the Kara Sea and Severnaya Zemlya in the West and the East Siberian Sea

and the New Siberian Islands in the East. The main sources of particulate organic carbon

(POC) for the Laptev Sea are terrigenous, both from coastal erosion and river runoff

(Sánchez-García et al., 2011; Stein and Macdonald, 2004). Marine primary production is

limited to on average two ice-free months per year and therefore generally low. Nutrient-poor

waters on the Siberian shelves resulting from a strong stratification further impede

phytoplankton growth (Sakshaug, 2004).

The destabilization of Pleistocene Ice-Complex Deposits along the coastline is a main

sediment source for the Laptev Sea (Rachold et al., 2000). The total POC input from coastal

erosion to Laptev and East Siberian Sea is estimated to be between 4.0 Tg yr$^{-1}$ (Semiletov,

1999; Stein and Fahl, 2000) and 22 ± 8 Tg yr$^{-1}$ (including net subsea permafrost-carbon

erosion, Vonk et al., 2012).

The Lena River is estimated to provide 20.7 Tg of sediment per year (Holmes et al., 2002),

i.e. > 70 % of the total riverine input to the Laptev Sea (Gordeev, 2006)with an average water

discharge of 588 km$^3$ yr$^{-1}$ (Holmes et al., 2012). It drains a watershed of ~2.46 x 10$^6$ km$^2$

(Holmes et al., 2012), of which 77 % is underlain by continuous permafrost (Amon et al.,

2012). Water discharge peaks in June, during the spring flood, when about 75 % of total

organic carbon is delivered (Rachold et al., 2004). Total POC discharge by the Lena River

can be up to 0.38 Tg yr$^{-1}$ (Semiletov et al., 2011).

Sediment transport pathways are largely influenced by the prevailing atmospheric conditions:

During cyclonic summers (i.e. positive phase of the Arctic Oscillation), northerly winds

predominate, strengthening the Siberian Coastal Current, which transports Lena River water

masses along the coast towards the East Siberian Sea; whereas during anticyclonic

summers (i.e. negative phase of the Arctic Oscillation and mainly southerly winds) the Lena

River plume is exported onto the mid-shelf and towards the deep part of the Arctic Ocean (Charkin et al., 2011; Dmitrenko et al., 2008; Guay et al., 2001; Wegner et al., 2013; Weingartner et al., 1999). Sediment transport in the Laptev Sea is strongly seasonal. The main transport of Lena River water with high concentrations of suspended particulate matter (SPM) towards the mid-shelf takes place shortly after river-ice breakup (Wegner et al., 2005). During the ice-free summer, SPM circulates between inner and mid-shelf with very little material escaping over the shelf break to the deeper parts of the Arctic Ocean. Significant sediment export is suggested to happen during freeze-up through SPM that is incorporated in sea ice and then transported across the continental margin (Dethleff, 2005; Eicken et al., 1997) or through the formation of dense bottom water resulting from brine ejection (Dethleff, 2010; Ivanov and Golovin, 2007). Hardly any sediment transport occurs beneath the ice cover.

Holocene-scale linear sedimentation rates for the Laptev Sea vary between 0.12 and 0.59 mm yr$^{-1}$ according to $^{14}$C dating of marine bivalves (Stein and Fahl, 2004, and citations therein), whereas centennial-scale $^{210}$Pb-derived rates for the more recent Laptev Sea can be up to 1.3 mm yr$^{-1}$ (Vonk et al., 2012). These rates do not seem to be correlated with water depth on the shelf, but values for the continental slope and rise tend to be on the lower end (0.12-0.38 mm yr$^{-1}$) (Stein and Fahl, 2004, and citations therein).

2.2  Sampling

Sediment sampling locations span from close to the Lena River mouth and in the Buor-Khaya Bay, across the shelf, to the continental slope and rise, covering a transect of  > 800 km with water depths increasing by more than two orders of magnitude. Samples SW-1, SW-2, SW-3, SW-4, SW-6, SW-14, SW-23 and SW-24 were collected during the SWERUS-C3 expedition on IB *ODEN* during summer 2014 using an Oktopus multicorer (8 Plexiglas tubes, 10 cm diameter). All other samples were collected during the International Siberian Shelf Study (ISSS-08) expedition onboard the RV *Yacob Smirnitskyi* during summer 2008. The YS-4, YS-6, YS-13 and YS-14 samples were taken with a GEMAX gravity corer (2 Plexiglas

tubes, 9 cm diameter); YS-9 and TB-46 were collected with a Van Veen grab sampler. For
the grab samples only surface sediments (uppermost cm) were subsampled and used in this
study. Sediment cores were cut into 1 cm slices within 24 hours after sampling. To account
for lower sediment accumulation rates on the rise, for SW-1, SW-2, SW-3 and SW-4 a higher
resolution of 0.5 cm for the top 10 cm was chosen. The depositional age for all samples is
thus between ~8 and ~70 years (depending on which sedimentation rates are employed). All
samples were kept frozen throughout the expedition and freeze-dried upon arrival to
Stockholm University laboratories. See Semiletov and Gustafsson (2009) for more
information on the ISSS-08 expedition. For exact sampling locations see Table 1.

2.3  Surface area
All surface area analyses have been performed on a Micromeritics Gemini VII Surface Area
and Porosity analyzer. Freeze-dried subsamples of ~0.7 g were furnaced at 400 °C for 12 h
and gently cooled down to room temperature to remove all organic material. Keil and Cowie
(1999) have shown that this method yields statistically similar results to the method using
removal with sodium pyrophosphate/ hydrogen peroxide (Mayer, 1994). The samples were
then desalted by repeated mixing with 50 ml of MilliQ water and centrifugation (20 min at
8000 rpm), followed by further freeze-drying. Directly prior to analysis they were degassed in
a Micromeritics FlowPrep 060 Sample Degas System for 2 h at 200 °C under a constant
nitrogen flow. Each analysis was initiated by measuring the free space in the vial. The
specific surface areas were derived from 6 pressure-point measurements (relative pressure
p/p0 = 0.05-0.3, equilibration time 5 s) with nitrogen as adsorbing gas (Brunauer et al., 1938).
The instrumental precision was 0.1-0.3 $m^2$ $g^{-1}$, which corresponds to a relative uncertainty of
about 1 %. The performance of the instrument was monitored with the surface area
reference material Carbon Black (21.0 ± 0.75 $m^2$ $g^{-1}$) provided by Micromeritics.

2.4 X-Ray Fluorescence
The mineral composition of ~1 g freeze-dried, homogenized subsamples was also
characterized with a wavelength dispersive sequential Philips PW2400 X-ray Fluorescence
(XRF) spectrometer. Prior to the analysis, sediment samples were combusted for 12h at 450
°C to remove the organic fraction. The XRF was operated under vacuum conditions on
samples prepared as glass beads using lithium tetraborate and melted with a fluxer Claisse
Fluxy (~1150°C) (Mercone et al., 2001). The relative error was less than 0.6 % for major
elements and less than 3 % for trace elements. In this study only $SiO_2$, $Al_2O_3$ and $CaO$ are
reported.

2.5 Bulk elemental and carbon isotope analysis
Concentration and $\delta^{13}C$ isotopic composition of total organic carbon (TOC) were determined
at the Stable Isotope Laboratory, Department of Geological Sciences, Stockholm University.
Homogenized subsamples of ~10 mg were repeatedly acidified (HCl,1.5 M, Ag capsules) to
remove carbonates (Nieuwenhuize et al., 1994). TOC concentrations and $\delta^{13}C$ isotopic
composition were measured simultaneously with a Carlo Erba NC2500 elemental analyzer
connected via a split interface to a Finnigan MAT Delta V mass spectrometer. TOC
concentrations were blank corrected and the relative error was < 1 %. Stable isotope data
are reported relative to VPDB using the $\delta^{13}C$ notation.
Radiocarbon analyses of acidified samples were conducted at the US National Ocean
Sciences Accelerator Mass Spectrometry (NOSAMS) Facility of the Woods Hole
Oceanographic Institution, USA, according to their standard routines (Pearson et al., 1998).
The relative error of the measurements was < 0.5 %. Radiocarbon data are reported using
the $\Delta^{14}C$ notation following Stuvier and Polach (1977).

2.6  Biomarkers
2.6.1 CuO-oxidation products
Microwave-assisted alkaline CuO oxidation was performed according to the method
established by Goñi and Montgomery (2000). Homogenized subsamples of 100-400 mg of
sediment (corresponding to 2-5 mg OC) were combined with 300 mg of copper(II) oxide and
50 mg of ferrous ammonium sulfate and oxidized under oxygen-free conditions (degassed
NaOH, 8 wt %) at 150 °C for 90 min using an UltraWAVE Milestone 215 Microwave oven.
After oxidation, known amounts of trans-cinnamic acid and ethyl vanillin were added as
recovery standards. Samples were acidified to pH 1 with HCl (12 M) and repeatedly
extracted with ethyl acetate. Anhydrous $Na_2SO_4$ was added to remove remaining water. The
solvent was evaporated and extracts re-dissolved in pyridine. For quantification, subsamples
were derivatized with bis-trimethylsilyl-trifluoroacetamide (BSTFA) + 1 %
trimethylchlorosilane (TMCS) and analyzed on a gas-chromatograph mass spectrometer in
full scan mode (GC-MS, Agilent) using a DB5-MS capillary column (60 m x 250 µm, 0.25 µm
stationary phase thickness, Agilent J&W) with a temperature profile of initially 60 °C followed
by a ramp of 5 °C min$^{-1}$ until reaching and holding 300 °C for 5 min. The quantification of
lignin phenols, benzoic acids, and p-hydroxybenzenes was achieved by comparison to the
response factors (key ions) of commercially available standards. For cutin-derived products,
fatty acids and dicarboxylic acids the response factor of trans-cinnamic acid was used as in
Goñi et al. (1998).

2.6.2 Solvent-extractable lipids
Wax lipids were extracted by means of accelerated solvent extraction (Dionex ASE 300)
using dichloromethane:methanol (9:1) according to the method described by Wiesenberg et
al. (2004). Pre-rinsed stainless-steel vessels were loaded with ~3 g of freeze-dried sediment,
filled up with pre-combusted glass beads and pre-combusted glass fiber filters at both ends.
Two extraction cycles were performed per sample applying a static pressure of 1500 psi and
a temperature of 80 °C for 5 min after a heating phase of 5 min. The flush volume was 50 %
of the 34 ml cell size with a purging time of 100 s.
Extracts were further purified (addition of activated Cu for sulfur and anhydrous $Na_2SO_4$ for
water removal) and then separated into a neutral and an acid fraction using BondElut
cartridges (bonded phase $NH_2$, Varian), eluting with dichloromethane:isopropanol (2:1) for
the neutral and methyl *tert*-butyl ether with 4 % acetic acid for the acid fraction according to
the method described by van Dongen et al. (2008a). The neutral fraction was further
separated into a polar and a non-polar fraction with an $Al_2O_3$ column. For each of the three
compound classes *n*-alkanes (neutral non-polar fraction), *n*-alkanols (neutral polar fraction)
and *n*-alkanoic acids (acid fraction) ~10 mg of one internal standard, $d_{50}$-tetracosane, 2-
hexadecanol and $d_{39}$-eicosanoic acid respectively, were added to the sediment samples prior
to extraction. All fractions were then analyzed on a GC–MS (Agilent) using the same column
and temperature program as for the CuO products. The polar and acid fractions were
derivatized with BSTFA + 1 % TMCS prior to analysis. Quantification was performed using a
5-point calibration curve with commercially available standards. Here, we only report data for
high-molecular weight (HMW) *n*-alkanes and *n*-alkanoic acids, where HMW refers to carbon
chain-lengths of ≥ 23 for *n*-alkanes and ≥ 24 for *n*-alkanoic acids.

## 3  Results and Discussion

The fate of permafrost-released terrigenous organic carbon (TerrOC) across the Laptev Sea shelf is controlled by competing processes. Degradation and sorting, as well as replacement of TerrOC by autochthonous marine organic matter all co-occur to varying degrees during cross-shelf transport. To disentangle their effects on the fate of permafrost-released TerrOC we first report changes in bulk sediment and OC properties and then focus on differences on the molecular level.

3.1 Characterization of the transect on a bulk level

Bulk total organic carbon (TOC) concentrations decreased across the shelf with highest values (~2 %) at shallow water depths and lowest values on the shelf edge (~0.8 %); at high water depths (> 2000 m) concentrations were slightly higher (~1 %). TOC values and the general pattern were in accordance with previous data from the Laptev Sea (Semiletov et al., 2005; Shakhova et al., 2015; Stein and Fahl, 2004; Vonk et al., 2012) and within the same range of those measured for the North American Arctic margin (Goni et al., 2013). Normalizing TOC concentrations to the mineral-specific surface area (SA) helps to understand the influence of physical sorting and preferential deposition on the observed TOC trends since SA is correlated to the sediment grain size to a first order approximation. To test if the mineral surface area is altered by the input of autochthonous organisms with siliceous or carbonaceous skeleton (e.g. silicoflagellates/diatoms or foraminifera/shells respectively), the mineral composition of the sediments was examined by X-ray fluorescence analysis. There were no apparent trends with water depth for either $SiO_2/Al_2O_3$ or $CaO/Al_2O_3$; therefore, marine production is not expected to have a measureable effect and SA can thus be regarded as a conservative parameter. This was also confirmed by low biogenic silica concentrations for the Laptev Sea reported earlier (< 1.4 %, Mammone, 1998). The relationship between TOC and SA has been widely studied on continental margins (e.g. Blair and Aller, 2012; Keil et al., 1994; Mayer, 1994). The TOC/SA ratios of typical river suspended sediments range between 0.4 and 1 mg m$^{-2}$ (Mayer, 1994). TOC/SA ratios > 1

mg m$^{-2}$ have been found in areas with high TOC supply (e.g. river outlets) and where the
deposited organic matter had spent little time under oxic conditions (short oxygen exposure
time, OET) (Mayer et al., 2002). Ratios < 0.4 mg m$^{-2}$ generally correspond to sediments from
deeper parts of the ocean and long OETs (e.g. Aller and Blair, 2006). Accordingly, the
TOC/SA values along the Laptev Sea transect displayed a strong decrease from 2.2 and 1.7
mg m$^{-2}$ close to the Lena River delta (water depths of 11 and 7 m, respectively) to about 0.3
mg m$^{-2}$ at water depths greater than 2000 m (Fig. 2A), proposing extensive TOC loss during
cross-shelf transport.
Bulk TOC isotopes have been broadly used to distinguish between organic matter sources.
Radiocarbon isotopes ($^{14}$C) convey information about the age of organic material, with
younger OC having higher $\Delta^{14}$C values. Marine organic matter produced primarily from $CO_2$
is expected to have modern $^{14}$C signatures, whereas permafrost-derived TerrOC has aged
both on land and during transport and has thus more depleted $^{14}$C values. The $\Delta^{14}$C values
for our Laptev Sea transect were generally low (< -280 ‰, Fig. 2B), suggesting a significant
input of pre-aged TerrOC (as in Vonk et al., 2012). Bulk TOC showed less depleted $\Delta^{14}$C
signatures with increasing distance from land on the shelf (from about -500 ‰ to about -340
‰ on the outer shelf, Fig. 2B), reflecting a dilution of older TerrOC with younger marine
material. On the slope and rise, however, $\Delta^{14}$C values decreased again to about -410 ‰.
This difference may be a result of ageing during lateral transport and/or after deposition due
to lower accumulation rates on slope and rise. The range between -340 ‰ and -410 ‰
corresponds to a $\Delta^{14}$C age difference of about 900 years; however, the depositional age
differences between shelf and slope samples were estimated to be less than 80 years (see
Section 2.2). Ageing after burial alone does therefore not explain the difference in $\Delta^{14}$C. Keil
et al. (2004) estimated a lateral transport time of 1800 years across the Washington margin
(158 km) from $\Delta^{14}$C data of bulk OC in surface sediments. For the > 200 km distance
between mid-shelf and rise a bulk ageing of 900 years does therefore not seem
unreasonable. It has to be taken into account, however, that mainly the TerrOC fraction of
the bulk OC is subject to such protracted lateral transport. Transport times would thus have
to be significantly higher in order to explain this age difference for the entire bulk OC. One
indication supporting this hypothesis of protracted lateral transport of TerrOC is the
degradation status of TerrOC at the deep stations. All molecular degradation proxies point
towards highly reworked material (see Section 3.3), suggesting that only the most refractory
TerrOC fraction is found at great water depths off the continental margin. Alternatively, the
lower $\Delta^{14}$C values at high water depths may be the consequence of more effective
degradation of marine organic matter throughout the water column, resulting in a
comparatively lower input of young autochthonous material. However, this latter scenario is
not supported by the stable carbon isotopic signature.
For stable carbon isotopes ($^{13}$C), terrigenous sources are generally more depleted than
marine organic matter (Fry and Sherr, 1984). In this study, values for $\delta^{13}$C of TOC ranged
between -26.5 ‰ and -22.3 ‰. The trend towards more enriched TOC with increasing
distance from the coast (Fig. 2B) can be explained by a growing proportion of marine organic
matter. However, the $\delta^{13}$C signature of the marine source appeared to be heavier than typical
marine planktonic material in that region (-26.7 ± 1.2 ‰, Panova et al., 2015; -24 ± 3 ‰,
Vonk et al., 2012, and citations therein). One possible explanation for this discrepancy is an
underestimated influence of ice algae that were reported to have highly enriched $\delta^{13}$C values
between -15 to -18 ‰ (Schubert and Calvert, 2001). Significant seafloor deposition of ice
algal biomass has been observed previously for the Arctic basins (Boetius et al., 2013).
Another option would be a more refractory, isotopically-enriched marine endmember (-21.2
‰) as suggested by Magen et al. (2010). They argue that lighter isotopes are preferentially
consumed by bacteria, which in turn enriches the remaining marine organic matter. Following
their reasoning, the more enriched values observed for this transect may be interpreted as
an increasing proportion of refractory marine organic matter.
Winterfeld et al. (2015b) analyzed surface water particulate organic carbon (POC) in the
Lena River delta and found a mean $\delta^{13}$C of -29.6 ± 1.5 ‰. Karlsson et al. (2011) reported
similarly depleted $\delta^{13}$C values for POC from the Buor-Khaya Bay (-29.0 ± 2.0 ‰), while their
mean value for sedimentary OC for the same stations was significantly more enriched (-25.9
± 0.4 ‰) and agreed well with our data for the shallow stations (-26.2 ± 0.3 ‰, stations YS-
13, YS-14 and TB-46). Lena River POC $\delta^{13}$C values from high-discharge periods agree well
with the more enriched values we found for the shallow stations (Rachold and Hubberten,
1998). Stein and Fahl (2004), Semiletov et al. (2011, 2012) and Vonk et al. (2012) presented
similar $\delta^{13}$C ranges and trends for sediments from parts of the Laptev Sea as is reported in
the current study for the entire width of  the Laptev Sea shelf. For the Arctic Amerasian
Continental shelf, Naidu et al. (2000) reported contrasts in absolute $\delta^{13}$C values comparing
surface sediment samples from different regions, but all commonly displayed an increasing
trend for $\delta^{13}$C values across the shelf, suggesting a growing fraction of marine organic matter
with increasing distance from the coast.
Combining TOC/SA ratios with stable isotope signatures ($\delta^{13}$C) may serve to disentangle two
different processes, which occur synchronously during cross-shelf transport (as in Keil et al.
1997a): 1.) The net loss of TerrOC and 2.) the replacement of TerrOC with autochthonous
marine OC. Net loss of TerrOC, caused by either degradation or hydrodynamic sorting during
transport, has been quantified previously using TOC/SA ratios (e.g. Aller and Blair, 2006; Keil
et al., 1997a). The carrying-capacity of inorganic particles for OC is assumed to be a function
of the SA (Mayer, 1994); a decrease in TOC/SA values can therefore be regarded as TOC
net loss.
Replacement of TerrOC with autochthonous marine OC does not change this ratio. But since
marine OC is known to be isotopically enriched in $\delta^{13}$C over TerrOC, this process is recorded
by an increasing isotopic signature. Along the Laptev Sea transect, both processes seemed
to play an important role (Fig. 2C). High TOC/SA values close to the Lena River decreased
sharply outbound in the nearshore regime, pointing to extensive net loss, while the increase
in $\delta^{13}$C values was minor in this area. Once TOC/SA ratios were < 0.8 mg m$^{-2}$ (water depths
> 20 m), the isotopic changes and thus the replacement of TerrOC with marine OC became
increasingly important. Similar trends were observed for the Amazon River delta (Keil et al.,
1997b).
However, the TOC/SA trend in the shallower sediments is likely driven by both degradation
of OC bound to the mineral matrix during cross-shelf transport and sorting of vascular plant
fragments that are retained in the inner shelf. A recent study (Tesi et al., 2016) has shown
that ~50 % of the total OC pool in the inner Laptev shelf surface sediments exists in the form
of large vascular plant fragments. They are trapped close to the coast due to their size and
resulting settling (Stoke's law), while the OC bound to the fine mineral matrix is more buoyant
and transported offshore towards deeper waters.

3.2  Molecular indicators of organic matter sources
3.2.1   Biomarker distributions
The abundances of different source-diagnostic molecular proxies have been extensively
investigated to elucidate complex carbon-cycling mechanisms. In this study, a biomarker
suite of CuO oxidation products and solvent-extractable lipids was analyzed in order to gain
more insights on TerrOC sources and degradation status along the Laptev Sea transect. All
biomarker concentrations were normalized to the sediment-specific surface area (SA)
instead of OC content to avoid the signals being overshadowed by other carbon pools. As
shown by the lack of water-depth-related changes in the mineral composition (Section 3.1),
mineral-matrix dilution by biogenic material is negligible.
Lignin-derived phenols have been widely used to trace TerrOC in the marine environment
(e.g. Ertel and Hedges, 1984; Goñi and Hedges, 1995; Hedges and Mann, 1979). The lignin
macro-molecule is only synthesized in vascular plants (and certain seaweed species that are
not existing in the study area) to render stability to the cell walls. Lignin-derived phenols are
typically grouped by phenol type (V: vanillyl phenols, i.e. vanillin, acetovanillone, and vanillic
acid; S: syringyl phenols, i.e. syringaldehyde, aceto syringone, and syringic acid; C: cinnamyl
phenols, i.e. p-coumaric and ferulic acids). Total lignin refers to the sum of the three groups.
Across the shelf, lignin loadings decreased substantially with increasing distance from the
coast/water depth (45 µg m$^{-2}$ close to the coast, 0.43 ± 0.09 µg m$^{-2}$ for the deep stations; loss
of 99.1 ± 0.2 %, Fig. 3A).
Cutin-derived hydroxy fatty acids are another compound class obtained from CuO oxidation,
which have been used in parallel with lignin phenols (e.g. Goñi et al., 2000; Prahl et al.,
1994). They are mainly associated with the soft tissues of vascular plants such as leaves and
needles. Cutin acid loadings displayed a similar trend as lignin phenols (11 µg m$^{-2}$ close to
the coast, 0.061 ± 0.010 µg m$^{-2}$ for the deep stations; loss of 99.4 ± 0.1 %, Fig. 3A).
Similar values and sharp declines with increasing distance from the coast for lignin and cutin
have been observed for the whole East Siberian Arctic Shelf (ESAS) (Tesi et al., 2014) (Fig.
4 for comparison of lignin phenol concentrations with literature values for different Arctic
margins). A recent study (Winterfeld et al., 2015a) for the Buor-Khaya Bay (5.8-17 m water
depth) reported lignin phenol concentrations on the same order of magnitude, up to 40 %
higher for the shallowest samples, and decreasing with increasing depth. For the Beaufort
Sea shelf, Goñi et al. (2000) found a less drastic decline in lignin phenols and cutin acids
going from 5 m water depth to 210 m, which likely reflected both lower concentrations in the
shallow waters (factor of ~2), and a narrower and steeper shelf. Lignin phenols were also
higher at greater water depths than on the ESAS. This may reflect the differences in
bathymetry: since the Beaufort Sea shelf is not as wide as, but steeper than, the ESAS,
lateral transport is possibly faster, leaving less time for organic matter to be degraded along
the way. A comparison between different shelf-slope systems across the North American
Arctic margin (Goni et al., 2013) revealed very low lignin and cutin concentrations for the
Canadian Archipelago, Lancaster Sound and Davis Strait, whereas both concentrations and
trends with water depth for the Beaufort Sea, Chuckchi Sea and Bering Sea were similar to
the results from this study. An exception to these patterns was Barrow Canyon, where at
water depths of > 2000 m lignin and cutin concentrations were as high as the ones observed
close to the Lena River delta, pointing to efficient rapid TerrOC transfer with comparably
short oxygen exposure times through this active canyon (Goni et al., 2013) (Fig. 4 and Fig.
S1).
Solvent extractable high-molecular weight (HMW) *n*-alkanes and *n*-alkanoic acids make up
the major part of epicuticular leaf waxes (Eglinton and Hamilton, 1967) and have been
broadly employed as TerrOC biomarkers (for the Arctic Ocean e.g. van Dongen et al., 2008;
Yunker et al., 1995, 2005). HMW wax lipids in this study also presented a decreasing trend
with increasing water depth/distance from the coast, but to a lesser extent than lignin phenols
or cutin acids (HMW *n*-alkanes: 1.1 µg m$^{-2}$ close to the coast, 0.12 ± 0.02 µg m$^{-2}$ for the deep
stations; HMW *n*-alkanoic acids: 12 µg m$^{-2}$ close to the coast, 0.42 ± 0.29 µg m$^{-2}$ for the deep
stations; loss of 89 ± 2 % and 96 ± 3 %, respectively, Fig. 3B).
Previous studies in the same area reported similar lipid biomarkers concentrations, which
confirm the magnitude of the decreasing trends with increasing water depth (Karlsson et al.,
2011; Vonk et al., 2010) (Fig. S1). HMW *n*-alkane concentrations in the Beaufort and the
Chuckchi Sea (Belicka et al., 2004; Yunker et al., 1993) are in accordance with the ones
measured on the ESAS, but the shallowest sample on the Beaufort Shelf is ~2 times lower
than the shallow ESAS samples (Fig. S1). This might imply that sediments transported by the
Mackenzie River to the Beaufort Shelf have lower TerrOC concentrations than Lena River
transported sediments. For the Mackenzie Shelf, Goñi et al. (2000) used lignin phenols and
cutin acids to estimate a terrigenous δ$^{13}$C endmember and therewith derived a terrigenous
contribution of almost 80 % for the shallowest sediments, while rough estimates from C/N
and δ$^{13}$C data suggested that TerrOC made up only 30-50 % of the organic carbon
(Macdonald et al., 2004). For the Lena Delta, source apportionment calculations using δ$^{13}$C
and Δ$^{14}$C data attributed up to 83 % of the organic carbon to terrigenous sources (Vonk et al.,

421     2012).

All TerrOC biomarker loadings displayed a strong decrease across the shelf, but their relative
losses differ substantially between compound classes (Fig. 3C). These findings agree with
previous results for the ESAS (Tesi et al., 2014), where similar differences between
biomarkers were reported. A somewhat larger decrease was observed for lignin than for
cutin, in contrast to this study. The different extents of biomarker losses for the different
compound classes may either be attributed to preferential degradation of lignin phenols and
cutin acids, implying that they are more labile than HMW *n*-alkanes and *n*-alkanoic acids, or
sorting during transport, suggesting that they are associated with a sediment fraction that is
hydraulically more retained and carried less efficiently to the outer shelf/slope. A recent study
(Tesi et al., 2016) aimed to disentangle these two processes by analyzing different fractions
of bulk surface sediments from three transects (yet with only three stations each) acrossthe
ESAS. The fractions were separated according to density (1.8 g cm$^{-3}$ cutoff), size (>63 µm,
38-63 µm, < 38 µm) and settling velocity (1 m d$^{-1}$ cutoff). The highest lignin phenol
abundance was found in low-density plant fragments (26-55 mg g$^{-1}$ OC). These large
particles have a higher settling velocity (Stokes' law) and are therefore hydraulically retained
close to the coast. Cutin acids and plant wax lipids were mainly associated with the high-
density fine (< 38 µm, > 1 m d$^{-1}$) and ultrafine (< 38 µm, < 1 m d$^{-1}$) mineral fractions. Within
the fine and ultrafine fractions, which made up about 95 % of the organic carbon on the outer
shelf, they found drastic losses of all biomarkers with increasing distance from the coast,
which they attributed to degradation during the protracted cross-shelf transport. Relative
decreases appeared to depend on the number of functional groups of the compound class:
98 ± 1 % for lignin phenols, 97 ± 1 % for cutin acids, 96 ± 1 % for HMW *n*-alkanoic acids and
89 ± 4 % for HMW *n*-alkanes. According to that study, the steep cross-shelf gradients
observed here for lignin phenols can be attributed to both hydrodynamic sorting close to the
coast and degradation during transport. From the data in the current study alone, the two
processes occurring in parallel - degradation and sorting during cross-shelf transport - cannot
be disentangled. However, using the data from (Tesi et al., 2016), we can make a rough
correction for the sorting part to derive an estimate of the net extent of degradation. For the
shallowest station in their study (same as here, TB-46), about 75 % of the lignin phenols
were associated with the low density fraction that was retained close to the coast. If we thus
assume only 25 %, i.e. 11 of the 45 µg m$^{-2}$ to be associated with the fine fraction that is
actually transported across the shelf, we obtain a reduction by 96 ± 1 % that can be
attributed to degradation (compared to 99.1 % net loss). These results agree with the values
presented in (Tesi et al., 2016). For the other compounds analyzed 55-74 % are associated
with the fine fraction even for the shallowest station and they therefore experience sorting to
a lesser extent.
Degradation after burial is assumed to play only a minor role. Differences in sedimentation
ages are expected to be small (Section 2.1) and a study on centennial-scale sediment cores
from the East Siberian Sea (Bröder et al., 2016) detected no significant TerrOC degradation
(as recorded by biomarker loss) with increasing sediment depth. Also in that study, lignin
phenol and cutin acid loadings were on average 20 times higher on the inner than on the
outer shelf, whereas for HMW *n*-alkanoic acids and *n*-alkanes the difference between inner
and outer shelf was only a factor of ~3-5. Contrasts between the stations were found to be
larger than down-core changes. This may be due to the fact that the cores in that study only
encompassed about one century of sedimentation ages, while the protracted cross-shelf
transport likely requires much longer timescales.

3.2.2 Lignin Phenol sources
Relative distributions of different lignin phenol classes reveal more information on TerrOC
sources since they are specific to different plant types. Syringyl phenols are not produced by
gymnosperm (non-flowering) plants; elevated syringyl to vanillyl ratios (i.e. S/V > 1, Hedges
and Parker, 1976) are therefore attributed to more lignin phenols from angiosperm
(flowering) plants. These ratios have to be handled with care, though, because the
preferential degradation of syringyl phenols by white- and brown-rot fungi on land can also
alter S/V ratios (Hedges et al., 1988). S/V values for the Laptev Sea transect increased with
increasing water depth from ~0.65 for the inner shelf to ~1.0 for the slope/rise sediments
(Fig. 5A). This trend can either be explained by preferential degradation of gymnosperm
material or sorting during transport. Tesi et al. (2014) measured generally lower values for
S/V (ESAS average: 0.47, for only Lena watershed dominated locations: 0.42) recording no
trend with water depth (Fig. S1 for comparisons with other studies). Their deepest station
was located at only 69 m water depth, though, whereas in this study sediments from down to
3146 m water depth were analyzed. S/V ratios in Buor-Khaya Bay surface sediments
(Winterfeld et al., 2015a) were also lower (0.43 ± 0.02 on average) and displayed no trend
with water depth. Within the water depth interval they studied (5.8-17 m), however, the

samples analyzed in this study had also quite homogeneous S/V ratios (0.64 ± 0.02). Two

sediment cores from the East Siberian Sea (Bröder et al., 2016) showed also lower S/V

values (inner shelf surface sediment: 0.62, outer shelf surface sediment: 0.50) displaying no

clear trends over time/down-core. For the Beaufort Sea shelf Goñi et al. (2000) detected

rather high values (0.54-1.71), which (besides the very high value at 61 m water depth)

agree with the data from this study. Other transects across the North American Arctic margin

(Goni et al., 2013) had slightly lower S/V ratios with no observed trends with water depth.

The ratio of cinnamyl to vanillyl phenols (C/V) is associated with the relative contributions of

woody versus soft material, because only non-woody vascular plants synthesize cinnamyl

phenols (Hedges and Mann, 1979a). This ratio admittedly decreases with ongoing

degradation (Opsahl and Benner, 1995) and may therefore not be used as an unambiguous

source indicator. We observed that C/V values strongly decreased across the Laptev Sea

Shelf from ~0.5 (close to the Lena River outlet) to ~0.1 (on the slope/rise, Fig. 5B), which

may reflect the preferential degradation of soft tissues. This trend is not likely caused by

hydrodynamic sorting, since typically the larger, low-density, woody plant fragments are

retained in shallower water, whereas finer material is transported further across the shelf

(e.g. Keil et al., 1994; Tesi et al., 2016).

C/V ratios in Buor-Khaya Bay sediments (Winterfeld et al., 2015a) in shallow waters were on

average lower and more homogeneous (0.17 ± 0.03) than those measured in this study (0.41

± 0.12 for the corresponding depth interval) (Fig. S1 for comparisons with other studies). C/V

values for the entire ESAS were on average 0.15 (0.14 ± 0.07 for only Lena dominated

waters) with no water depth trend (Tesi et al., 2014). In shallow sediment cores from the East

Siberian Sea, Bröder et al. (2016) measured C/V ratios of 0.20 (inner shelf) and 0.13 (outer

shelf) for the surface sediments with no significant trend over sediment depth. For the

Mackenzie Shelf C/V values ranged between 0.16 and 0.32 and slightly increased with

increasing water depth (Goñi et al., 2000). In contrast, in the Bering Sea, Chuckchi Sea,

Barrow Canyon, Canadian Archipelago, Lancaster sound and Davis Strait there were no C/V

trends observed (Goni et al., 2013), with lower values in the Canadian part (0.10 ± 0.12) and

highest values on the Beaufort Sea slope, where values slightly decreased with increasing
depth (0.39 ± 0.07).
A comparison to the S/V-C/V signatures of potential Arctic plant end-members (compiled by
Amon et al., 2012, and citations therein, Tesi et al., 2014, and Winterfeld et al., 2015a)
showed that lignin phenols likely derive from both angio- and gymnosperm soft tissues in the
shallower samples, closely matching with willow (*Salix*) tissues measured by Winterfeld et al.
(2015a). With increasing water depths, angiosperm wood became the most important source
material, while gymnosperm wood, grasses and mosses did not appear to contribute
significantly to the overall lignin phenol fingerprint (Fig. 5C). As discussed earlier, this trend
may well be a result of preferential degradation and sorting during cross-shelf transport and
not derive from actual changes in source material.

3.3 Degradation status of organic matter
During degradation, syringyl and vanillyl phenol aldehydes are oxidized to carboxylic acids of
the same phenol group. Increasing Sd/Sl and Vd/Vl ratios can therefore qualitatively indicate
ongoing degradation of lignin phenols (Ertel and Hedges, 1984; Hedges et al., 1988). For
fresh plant material typical acid-to-aldehyde ratios are around 0.1-0.2 (Hedges et al., 1988).
Winterfeld et al. (2015a), however, found values as high as Sd/Sl = 0.80 and Vd/Vl = 0.67 for
a moss species (*Aulacomnium turgidum*), Sd/Sl = 0.87 for larch (*Larix*) needles and Sd/Sl =
0.49 Vd/Vl = 0.41 for wild rosemary (*Ledum palustre*). Sedges (*Carex spp.*), dwarf birch
(*Betula nana*) and willow (*Salix*) range between Sd/Sl = 0.13-0.24 and Vd/Vl = 0.18-0.23.
The ratio of CuO oxidation-derived 3,5-dihydroxybenzoic acid to vanillyl phenols (3,5-Bd/V)
also serves as a proxy for degradation as 3,5-Bd is formed during humification likely
occurring in soils (Gordon and Goñi, 2004; Hedges et al., 1988; Prahl et al., 1994; Tesi et al.,
2014). For this reason, this proxy can trace mineral rich soil organic matter in contrast to
vascular plant debris (e.g. Dickens et al., 2007; Prahl et al., 1994) as well as degradation
during cross shelf transport (Tesi et al., 2016).
Sd/Sl, Vd/Vl and 3,5-Bd/V all increased along the transect, implying more degraded material
with increasing residence time in the shelf system (Fig. 6A). There appeared to be no
differences between outer shelf/slope and rise, which may indicate that TerrOC on the slope
is already highly reworked. In contrast, Tesi et al. (2014) found no correlation between Sd/Sl
or Vd/Vl and distance from the coast, while 3,5-Bd/V significantly increased with increasing
distance from the coast (Fig. S2 for comparisons with other studies). Sd/Sl values for the
Buor-Khaya Bay from Winterfeld et al. (2015a) were slightly higher (1.04 ± 0.24) than
samples from the corresponding water depths in this study (0.66 ± 0.15), whereas Vd/Vl
values were significantly higher (1.28 ± 0.30 compared to 0.59 ± 0.14). Measurements for the
Mackenzie Shelf agreed with the ones in this study (Sd/Sl = 0.81 ± 0.25 compared to 1.01 ±
0.33 for the corresponding water depths; Vd/Vl = 0.69 ± 0.14 to 0.86 ± 0.26; 3,5-Bd/V = 0.19
± 0.04 to 0.31 ± 0.15), but did not show a trend with water depth (Goñi et al., 2000).
Tesi et al. (2016) observed lower acid/aldehyde ratios for the lignin-rich low-density fraction
compared to the other fractions (high-density with different grain sizes and settling velocities)
in coastal surface sediments from the ESAS. With increasing distance from the coast, these
values increased, whereas for the other fractions there were no apparent trends. These
findings were interpreted as relatively fresh lignin in the low-density fraction (rich in large
plant fragments) compared to the relatively degraded lignin that had likely experienced
leaching and adsorbed to the fine mineral fractions (i.e. mineral bound OC). Bulk 3,5-Bd/V
values are potentially affected by both sorting and degradation, as they increased with
decreasing particle size (fine and ultrafine fractions had the most degraded signal and are
preferentially transported to the outer shelf) and across the shelf in each of the fractions.
The carbon preference indices for HMW *n*-alkanes and HMW *n*-alkanoic acids have also
been widely applied as degradation proxies for plant waxes in marine sediments (for the
ESAS, e.g. van Dongen et al., 2008; Fahl and Stein, 1997; Fernandes and Sicre, 2000; Vonk
et al., 2010). It measures the ratio of odd-to-even numbers of carbon chain-lengths of HMW
lipids and is based on the preference of odd carbon chain-lengths for HMW *n*-alkanes in
fresh plant material (even carbon chain-lengths for HMW *n*-alkanoic acids; Eglinton and
Hamilton, 1967). With ongoing degradation this preference is lost and the CPI approaches 1
(Bray and Evans, 1961).
We observed that the HMW $n$-alkane CPI presented  a similar pattern as the lignin phenol
based degradation indices. However, the HMW $n$-alkanoic acid CPI did not show as much of
a degradation trend (HMW $n$-alkane CPI: ~5.7 close to the coast, ~2.2 for the deep stations;
HMW $n$-alkanoic acids: ~5.4 close to the coast, ~4.1 for the deep stations; Fig. 6B). Karlsson
et al. (2011) measured lipid CPIs in the Buor-Khaya Bay with 10-80 km distance to the coast
and obtained similar results to this ~800 km cross-shelf study, with higher values closer to
the river delta (Fig. S2 for comparisons with other studies). Their data appears to have a
wider spread, though, which might be due to either the narrower dynamic range. Fahl and
Stein (1997) also reported a large range of $n$-alkane CPI values (< 0.2- > 5) for Laptev Sea
sediments. Fernandes and Sicre (2000) analyzed sediments from the Kara Sea and from the
major rivers discharging into this sea, Ob and Yenisey rivers. In the marine environment and
the Ob River, they observed HMW $n$-alkane CPI values between 4.8 and 5.3, similar to those
found at shallow water depths in this study. For the Yenisey River and mixing zone, they
found higher CPI values, pointing to fresher material being transported there. Vonk et al.
(2010) recorded HMW $n$-alkane CPI values for sediments along the East Siberian Sea
Kolyma paleoriver transect (across the East Siberian Sea) shelf that decreased from > 7.5 to
< 4.0 with increasing distance from the river mouth, overall higher than in this study but
confirming the general trend to more degraded material on the outer shelf. Tesi et al. (2016)
found HMW $n$-alkanoic acid CPI values to decrease with decreasing particle size with no
significant trends across the shelf in all but the low-density fraction, which is largely retained
close to the shore. The HMW $n$-alkane CPI values in that study, however, showed no
systematical differences between different fractions, but an overall decreasing trend with
increasing distance from the coast.
When undergoing degradation, HMW $n$-alkanoic acids may also lose their functional groups,
turning them into HMW $n$-alkanes (Meyers and Ishiwatari, 1993). The slightly decreasing
ratio of HMW $n$-alkanoic acids to $n$-alkanes also hints at more degraded material with
increasing water depth, although, due to a rather large variability, this trend is not significant.
For the Buor-Khaya Bay surface sediments Karlsson et al. (2011) obtained similar results
(0.48-10.7, here 1.1-10.9) with higher values closer to the river delta (Fig. S2 for
comparisons with other studies). Along the Kolyma paleoriver transect, Vonk et al. (2010)
measured HMW *n*-alkanoic acid to *n*-alkane ratios between 1 and 6 with no clear trend with
increasing distance from the river mouth. Tesi et al. (2016) found decreasing values with
increasing distance from the coast with no differences between the fractions. Two sediment
cores from inner and outer East Siberian Sea recording about one century of sedimentation
showed no clear trend in CPI or HMW *n*-alkanoic acid/*n*-alkane towards more degraded
TerrOC with increasing sediment depth (Bröder et al., 2016), but displayed a similar
difference between inner and outer shelf as seen in this study. This contrasting behavior for
cross-shelf and down-core trends may be caused by significantly different timescales for the
two processes: about one century in situ/after burial compared to potentially several millennia
long lateral transport. Furthermore, the degradation efficiency is likely higher under the oxic
conditions prevailing during cross-shelf lateral transport (Keil et al., 2004), than in the anoxic
conditions that predominate below a few millimeters of sediments on the ESAS (e.g. Boetius
and Damm, 1998). Comparing in situ to transport-related oxygen exposure times on the wide
Arctic shelves could potentially resolve the observed discrepancies.

## 4 Concluding remarks and future research directions

Across the Laptev Sea from the Lena River mouth to the deep sea of the Arctic interior a considerable loss of terrigenous organic matter has been observed on both bulk and molecular level. All terrigenous biomarkers display a massive decline with increasing water depth along this high-resolution transect due to hydrodynamic sorting and degradation during transport. Terrigenous organic matter (TerrOC) seems to be also qualitatively more degraded on the outer shelf, slope and rise compared to inner shelf and coastal areas.

These results corroborate and expand previous findings for the East Siberian Arctic Shelf, showing that the shelf seas in this region function as an active reactor for TerrOC. Since the East Siberian Arctic Shelf belongs to the widest and shallowest continental margins on Earth, cross-shelf transport times and thus the time spent in oxic sediments are expected to be comparatively long. This stands in contrast to e.g. the Mackenzie basin, which is thought to act as a geological sink for organic carbon due to its TerrOC burial (Hilton et al., 2015). For narrower Arctic shelves in general, where transport times can be expected to be much shorter, organic matter transfer towards the deeper basins appears to be much more efficient, with high TerrOC concentrations in surface sediments even at greater water depths (e.g. Barrow Canyon, Goni et al., 2013). It can therefore be assumed that the cross-shelf transport time exerts first-order control over the extent of TerrOC degradation. With ongoing global warming, rising permafrost-derived organic carbon input from river-sediment discharge and coastal erosion is expected to reach the marine environment. It is therefore crucial to better constrain cross-shelf transport times in order to determine a TerrOC degradation rate and thereby contribute to quantifying potential carbon-climate feedbacks.

**Acknowledgements**

We thank crew and personnel of the IB *ODEN*, the RV *Yakob Smirnitskyi* and the *TB0012*. The SWERUS-C3 and the International Siberian Shelf Study 2008 (ISSS-08) expeditions were supported by the Knut and Alice Wallenberg Foundation, Headquarters of the Far

Eastern Branch of the Russian Academy of Sciences, the Swedish Research Council (VR
Contract No. 621-2004-4039, 621-2007-4631 and 621-2013-5297), the US National Oceanic
and Atmospheric Administration (OAR Climate Program Office, NA08OAR4600758/Siberian
Shelf Study), the Russian Foundation of Basic Research RFFI (08-05-13572, 08-05-00191-a,
and 07-05-00050a), the Swedish Polar Research Secretariat, the Nordic Council of Ministers
and the US National Science Foundation (OPP ARC 0909546). L. Bröder also acknowledges
financial support from the Climate Research School of the Bolin Climate Research Centre. T.
Tesi also acknowledges EU financial support as a Marie Curie fellow (contract no. PIEF-GA-
2011-300259), contribution no. XXXX of ISMAR-CNR Sede di Bologna. J.A. Salvadó also
acknowledges EU financial support as a Marie Curie grant (FP7-PEOPLE-2012-IEF; project
328049). I. Semiletov thanks the Russian Government for financial support (mega-grant
#14.Z50.31.0012). O. Dudarev thanks the Russian Science Foundation (grant No. 15-17-

654    20032).

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

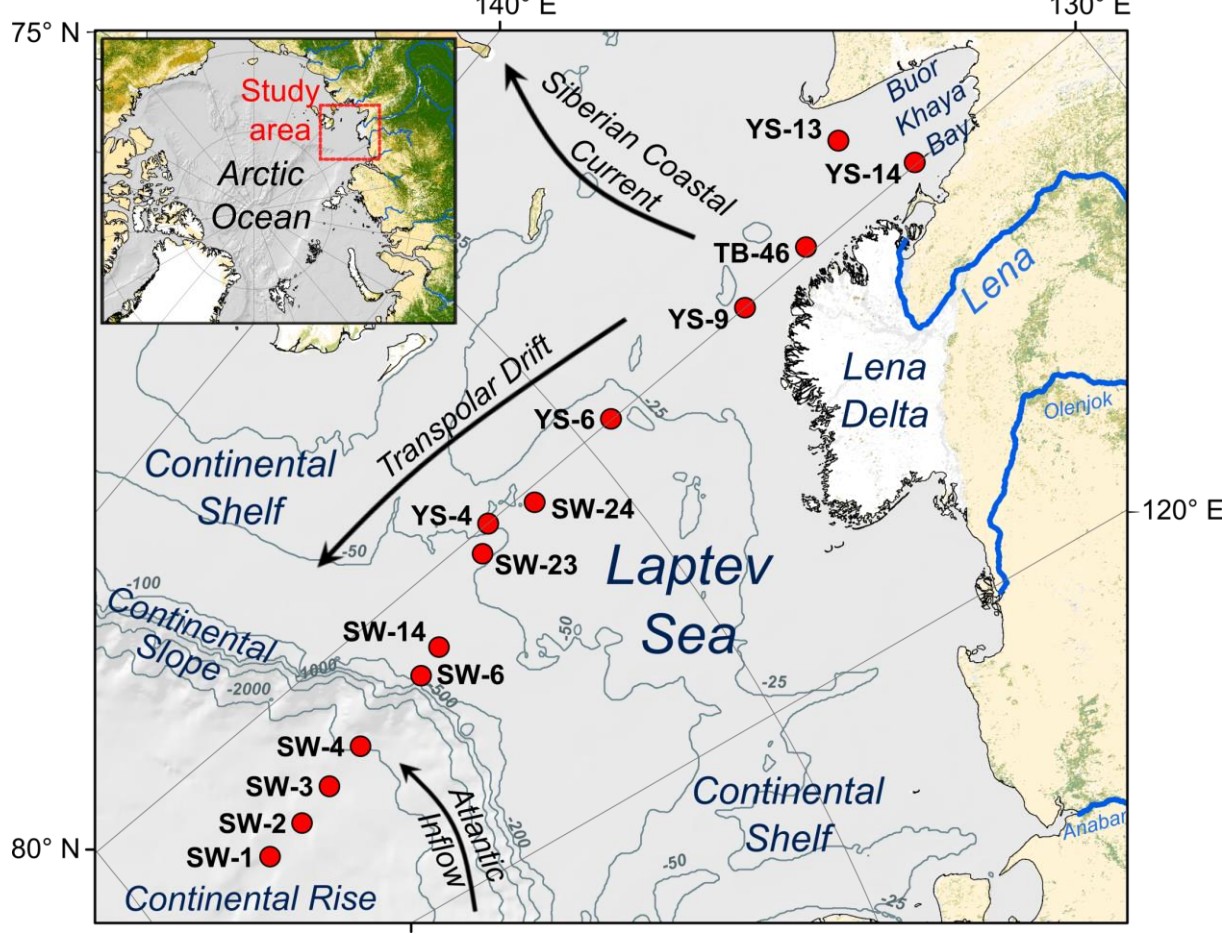


Figure 1: Map of the study area in the Laptev Sea. Red filled circles mark the sediment
sampling sites. The transect reaches from the Lena River mouth and the Buor-Khaya Bay
(water depths ~10 m) across the Laptev Sea Shelf (mean depth ~50 m) to the slope/shelf
break and rise (water depths ~3000 m). Arrows show the directions of the prevailing ocean
currents.

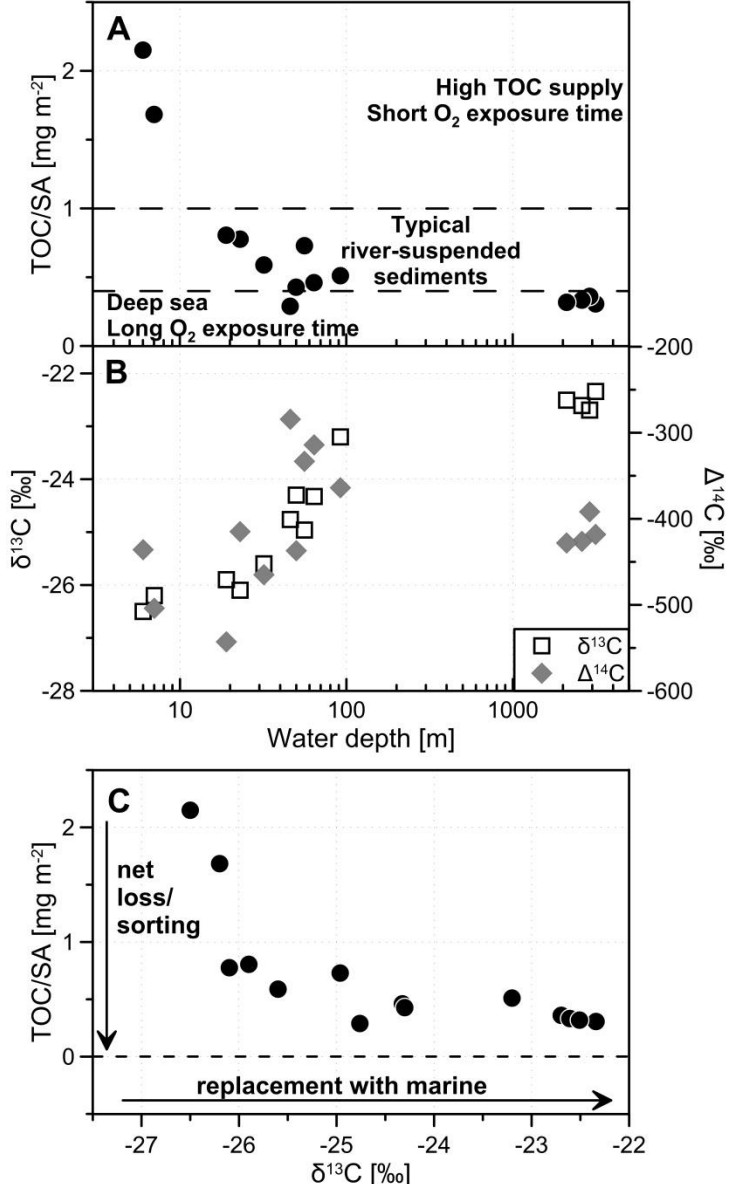


Figure 2: (A) The ratio of total organic carbon (TOC) to mineral surface area (SA). Typical
values for deep sea, river-suspended sediments and high TOC supply are taken from Blair
and Aller (2012). (B) The stable carbon isotopic signal ($\delta^{13}C$, open boxes) and the
radiocarbon isotopic signal ($\Delta^{14}C$, filled diamonds). (C) The relationship between TOC/SA
and $\delta^{13}C$ can help to disentangle two processes occurring simultaneously during cross-shelf
transport: The net loss (i.e. degradation) or sorting (i.e. hydraulically retaining) of TerrOC
leads to a shift towards lower TOC/SA ratios, whereas the replacement/dilution with marine
OC shifts the isotopic signature towards higher values.

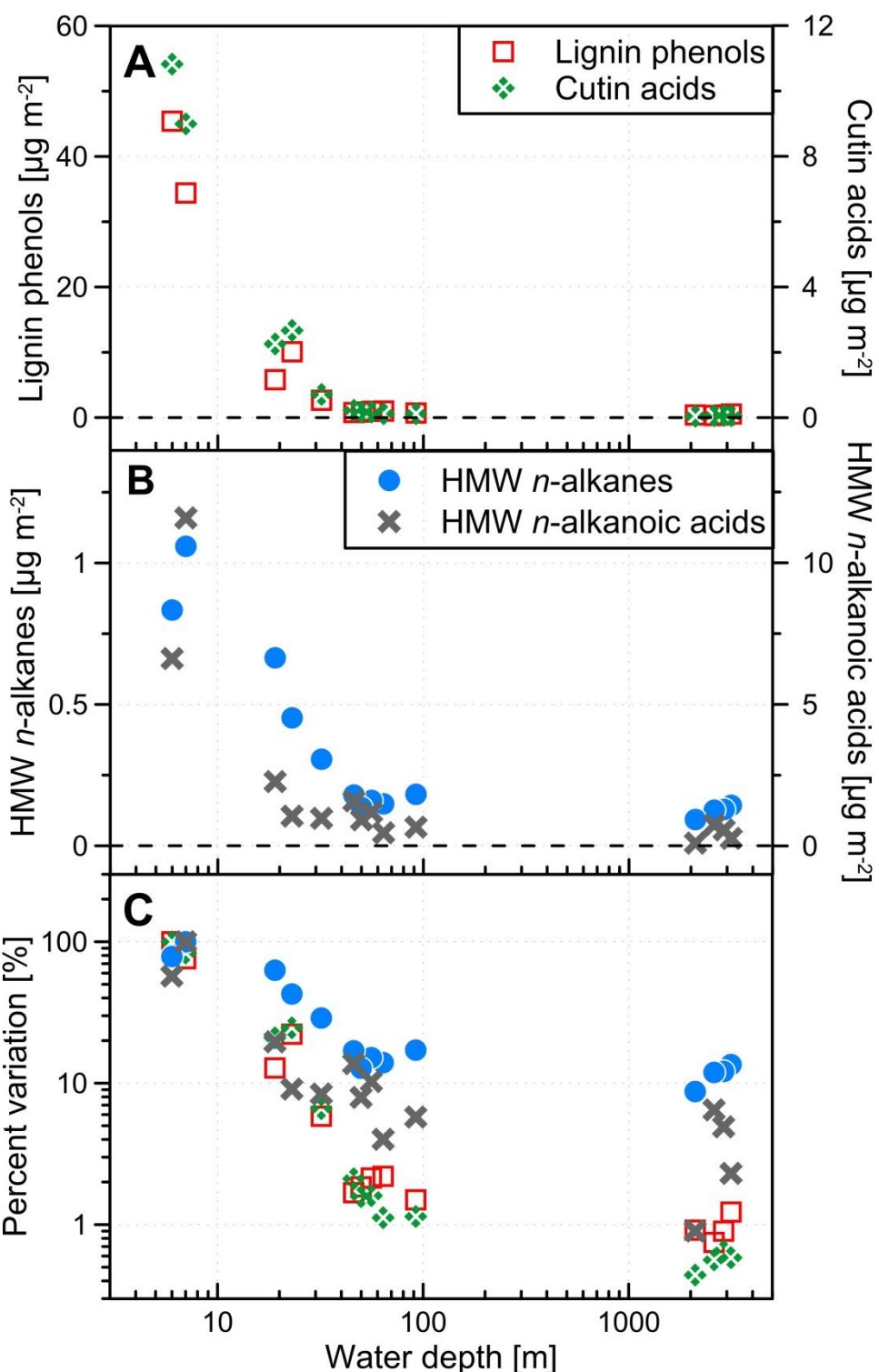


Figure 3: Terrigenous biomarker loadings across the shelf: (A) lignin phenols and cutin acids,
(B) HMW *n*-alkanes and HMW *n*-alkanoic acids. (C) Comparison between the different
biomarkers along the transect: lignin phenols, cutin acids, HMW *n*-alkanoic acids and *n*-
alkanes where each is normalized to respective highest value (corresponding to 100 %).

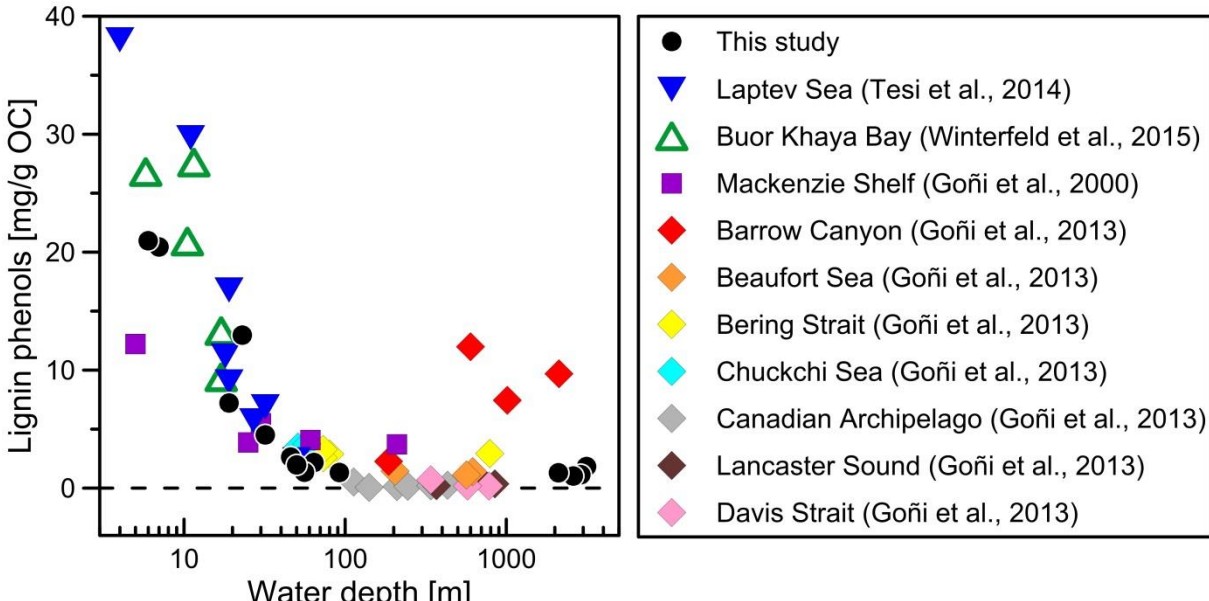


Figure 4: A comparison of lignin phenol data from this project to values from published
studies around the Arctic Ocean. Similar decreasing trends with increasing water depth are
observed for all systems but Barrow Canyon, where elevated lignin phenols concentrations
are found even at depth of > 1000 m.

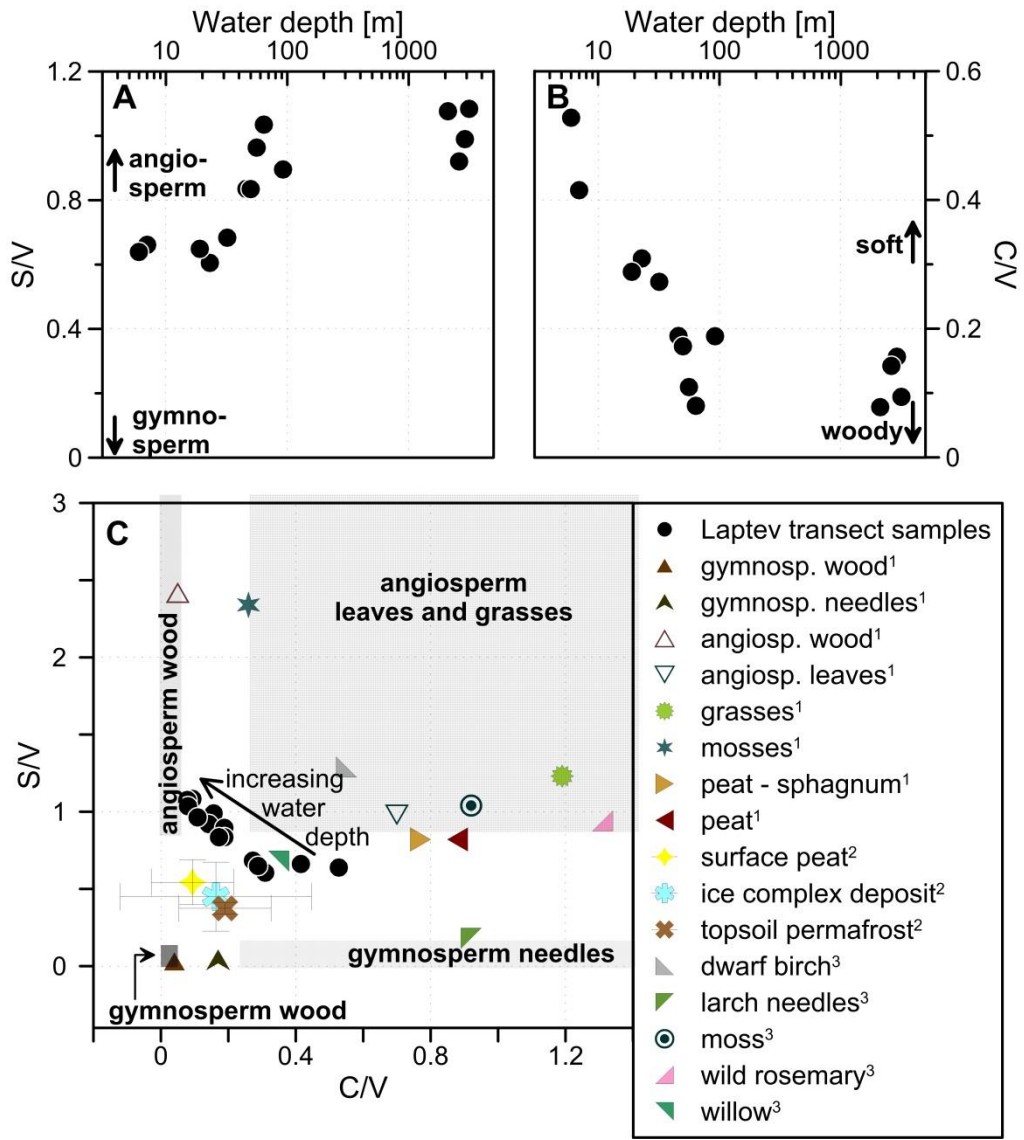


Figure 5: The lignin phenol composition carries source information: (A) an increasing ratio of syringyl to vanillyl phenols (S/V) suggests relatively more angiosperm material. (B) A decreasing ratio of cinnamyl to vanillyl phenols (C/V) implies an increasing relative contribution of woody material compared to soft tissues. (C) Comparison of S/V and C/V with the end-members for different Arctic plants as compiled from different studies by Amon et al. (2012, and citations therein, here marked with [1]); ice-complex deposit and topsoil permafrost as determined by Tesi et al. (2014, here marked with [2]) and more plant species measured by Winterfeld et al. (2015a, here marked with [3]). The boxes indicate typical ranges of S/V and C/V for different vascular plant tissues in different locations (e.g. Goñi et al., 2000).

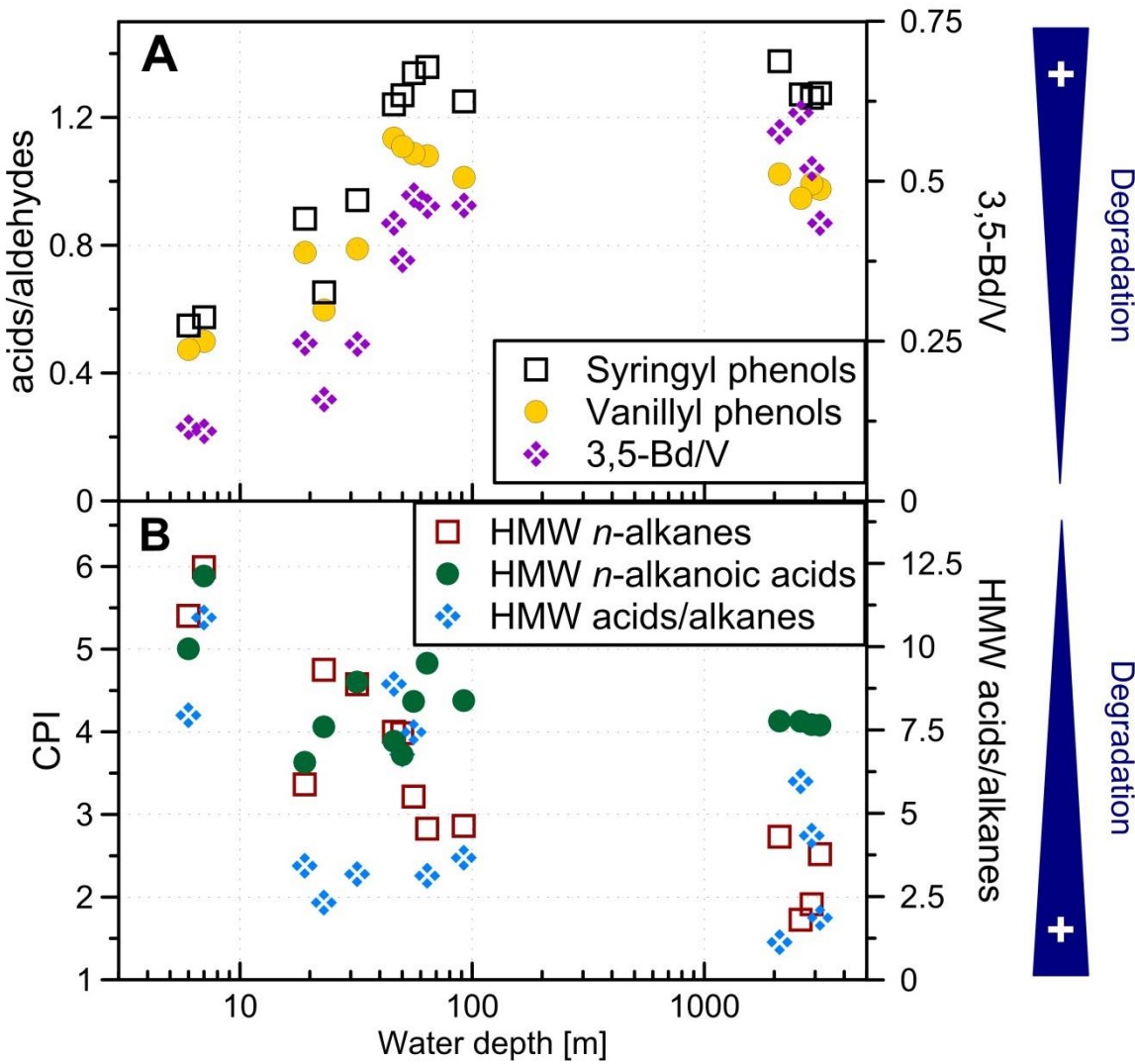


Figure 6: Degradation proxies for TerrOC, blue triangles point toward lower extent of
degradation: (A) CuO-oxidation derived ratios Sd/Sl, Vd/Vl and 3,5-Bd/V. (B) Carbon
preference indices (CPI) of HMW *n*-alkanes and *n*-alkanoic acidsand the ratio of HMW *n*-
alkanoic acids to HMW *n*-alkanes.

**Table 1: List of surface sediment samples from the Laptev Sea transect**

| ID | Sample type | Lat | Long | Water depth | OC | SA | $\delta^{13}C$ | $\Delta^{14}C$ | $SiO_2$ | $Al_2O_3$ | CaO |
|---|---|---|---|---|---|---|---|---|---|---|---|
| | | °N | °E | m | mg g$^{-1}$ | m$^2$ g$^{-1}$ | ‰ | ‰ | wt % | wt % | wt % |
| SW-1 | 0-0.5cm | 78.942 | 125.243 | 3146 | 10.4 | 34.0 | -22.34 | -418 | 60.3 | 16.5 | 2.4 |
| SW-2 | 0-0.5cm | 78.581 | 125.607 | 2900 | 13.8 | 38.3 | -22.70 | -392 | 57.8 | 17.2 | 2.1 |
| SW-3 | 0-0.5cm | 78.238 | 126.150 | 2601 | 10.6 | 31.8 | -22.61 | -426 | 62.1 | 16.0 | 1.6 |
| SW-4 | 0-0.5cm | 77.855 | 126.664 | 2106 | 13.2 | 41.5 | -22.51 | -428 | 56.6 | 17.5 | 1.3 |
| SW-6 | 0-1cm | 77.142 | 127.378 | 92 | 7.6 | 14.9 | -23.20 | -364 | 72.0 | 12.6 | 1.7 |
| SW-14 | 0-1cm | 76.894 | 127.798 | 64 | 8.9 | 19.4 | -24.33 | -314 | 71.3 | 12.5 | 1.5 |
| SW-23 | 0-1cm | 76.171 | 129.333 | 56 | 15.8 | 21.7 | -24.96 | -333 | 68.9 | 13.6 | 1.4 |
| YS-4 | 0-1cm | 75.987 | 129.984 | 50 | 13.4[a] | 31.4 | -24.76[a] | -284[a] | 63.8 | 15.1 | 1.3 |
| SW-24 | 0-1cm | 75.599 | 129.558 | 46 | 10.7 | 37.0 | -24.30 | -437 | 62.5 | 15.4 | 1.2 |
| YS-6 | 0-1cm | 74.724 | 130.016 | 32 | 18.6[a] | 31.6 | -25.60[a] | -465[a] | 62.1 | 16.1 | 1.3 |
| YS-9 | Grab | 73.366 | 129.997 | 23 | 13.1[b] | 16.9 | -26.10[b] | -415[b] | 70.8 | 14.0 | 1.3 |
| YS-13 | 0-1cm | 71.968 | 131.701 | 19 | 18.9[a] | 23.5 | -25.90[a] | -543[a] | 61.6 | 17.4 | 0.8 |
| YS-14 | 0-1cm | 71.630 | 130.050 | 7 | 19.1[a] | 11.4 | -26.20[a] | -504[a] | 69.6 | 15.0 | 1.6 |
| TB-46 | Grab | 72.700 | 130.180 | 6 | 25.8[a] | 12.0[c] | -26.50[a] | -436[a] | 67.9 | 15.2 | 1.8 |


[a] Data from Vonk et al. (2012); [b] data from Tesi et al. (2016); [c] data from Karlsson et al.
987 (2014).

**Table 2: Biomarker results for surface sediment samples from the Laptev Sea transect**

| ID | Lignin | Cutin | HMW* alkanes | HMW** acids | S/V | C/V | Sd/Sl | Vd/Vl | 3,5Bd/V | CPI alk | CPI acids | acids/ alk |
|----|--------|-------|--------------|-------------|-----|-----|-------|-------|---------|---------|-----------|-----------|
| | μg m⁻² | μg m⁻² | μg m⁻² | μg m⁻² | | | | | | | | |
| SW-1 | 0.56 | 0.063 | 0.14 | 0.27 | 1.1 | 0.09 | 1.3 | 0.98 | 0.43 | 2.5 | 4.1 | 1.9 |
| SW-2 | 0.41 | 0.070 | 0.13 | 0.57 | 0.99 | 0.16 | 1.3 | 0.99 | 0.52 | 1.9 | 4.1 | 4.3 |
| SW-3 | 0.34 | 0.061 | 0.13 | 0.75 | 0.92 | 0.14 | 1.3 | 0.95 | 0.61 | 1.7 | 4.1 | 6.0 |
| SW-4 | 0.42 | 0.048 | 0.093 | 0.10 | 1.1 | 0.08 | 1.4 | 1.0 | 0.58 | 2.7 | 4.1 | 1.1 |
| SW-6 | 0.68 | 0.12 | 0.18 | 0.67 | 0.90 | 0.19 | 1.2 | 1.0 | 0.46 | 2.9 | 4.4 | 3.7 |
| SW-14 | 1.0 | 0.12 | 0.15 | 0.46 | 1.0 | 0.08 | 1.4 | 1.1 | 0.46 | 2.8 | 4.8 | 3.1 |
| SW-23 | 0.97 | 0.17 | 0.16 | 1.2 | 0.96 | 0.11 | 1.3 | 1.1 | 0.48 | 3.2 | 4.4 | 7.4 |
| YS-4 | 0.84 | 0.17 | 0.13 | 0.92 | 0.83 | 0.17 | 1.3 | 1.1 | 0.38 | 4.0 | 3.7 | 6.8 |
| SW-24 | 0.76 | 0.23 | 0.18 | 1.6 | 0.84 | 0.19 | 1.2 | 1.1 | 0.43 | 4.0 | 3.9 | 8.9 |
| YS-6 | 2.7 | 0.71 | 0.31 | 0.97 | 0.68 | 0.27 | 0.94 | 0.79 | 0.25 | 4.6 | 4.6 | 3.2 |
| YS-9 | 10 | 2.7 | 0.45 | 1.1 | 0.60 | 0.31 | 0.65 | 0.60 | 0.16 | 4.7 | 4.1 | 2.3 |
| YS-13 | 5.8 | 2.3 | 0.64 | 2.3 | 0.65 | 0.29 | 0.88 | 0.78 | 0.25 | 3.4 | 3.6 | 3.4 |
| YS-14 | 34 | 9.0 | 1.1 | 12 | 0.66 | 0.42 | 0.57 | 0.50 | 0.11 | 6.0 | 5.9 | 11 |
| TB-46 | 45 | 11 | 0.83[d] | 6.6[d] | 0.64 | 0.53 | 0.55 | 0.47 | 0.12 | 5.4[d] | 5.0[d] | 7.9[d] |

 * carbon chain-lengths 23-34; ** carbon chain-lengths 24-30.

 [d] recalculated data from Karlsson et al. (2011).