# Peer review of "Fate of terrigenous organic matter across the Laptev Sea"

_Biogeosciences, 2016_

## Referee Comment (RC1) · X. Feng (Referee) · 24 May 2016

Degradation of terrestrial organic matter (TerrOC) along its transport into the ocean is an extensively investigated yet not fully understood aspect of the global carbon cycle. The Siberian Arctic Shelf, with a width > 800 km, is a unique and ideal place to study the transformation of TerrOC upon its entry into the sea. In particular, with climate-induced mobilization of permafrost-locked TerrOC, this area is receiving greater attention than many other shelves on earth. This paper uses a series of terrestrial biomarkers (including lignin phenols, cutin acids, and wax lipids) to study the abundances as well as degradation of TerrOC along a 800 km transect from Lena River mouth across the shelf to the slope and rise. In conjunction with carbon isotope and surface area analyses,

the authors display an increasing TerrOC degradation with increasing distance from the coast. The dataset is large and unique, and the writing is clear and organized. I have a few suggestions for the authors to consider.

First, in the Results and Discussion, many comparisons are made to other published Arctic (or other non-Arctic) studies, which is great and necessary. But the text is a bit reiterative and I wonder if there is a better way to display all the information from literature with tables or figures more vividly, which will help the readers to digest.

Second, TerrOC degradation is not unique to permafrost-derived OC: it happens in other shelf environments without permafrost OC input. What I am interested in this study is what is special about the transformation of TerrOC in the Siberian Arctic Shelf in comparison with other parts of the world or other depositional environment. I think the authors have made some very nice comparisons with the Mackenzie Shelf. But I think this may be more emphasized in the conclusions, etc.

Other minor points: L19: Change to "Mobilized permafrost carbon" can be either... L147: How does combustion affect surface area measurement? L165: What about FeOx, which plays a key role in the preservation of TerrOC? L346-348: How does hydrodynamic sorting affect the SA-normalized abundance of lignin? This is probably an important aspect (if not more important) other than degradation, which may explain the varied decrease rate for lignin vs. wax lipids. Does this bias the SA-normalized abundances?

---

## Referee Comment (RC2) · Anonymous Referee #2 · 18 Jul 2016

Review of Broder et al. "Fate of terrigenous organic matter across the Laptev Sea from the mouth of the Lena River to the deep sea of the Arctic interior"

Accept with major revisions

Pros
- Does fill a much needed role as it is one of few papers that looks at the fate of terrigenous organic matter as it is carried out past the continental shelf
- They do compare their data to other shelf studies but some of there comparisons are not valid (i.e. explaining differences in HMW degradation in different studies is due to differences in chain length)
- Seems likely that the amount of time spent during cross-shelf transport is correlated with terrigenous organic matter degradation

Cons
- Figure 1 should include coastal currents and could have an inset of where the Laptev Sea is relative to the rest of the arctic
- Need to justify in the paper that the terrigenous matter in this study is only coming from the Lena River and not from the two other rivers (shown in Figure 1 and are not labeled) that empty into the Laptev Sea
- Need specific references when discussing what TOC/SA ratios are expected for what kind of environment (i.e. river, deep ocean; lines 248-253).
- Line 272: Uses a lateral transport time for an active margin instead of one of a passive margin. Suggest using a east coast system from the U.S.
- Along the same lines as bullet point number 2 in this section, if you cannot prove the source of this OC is the same, then you can not prove that it is aging
- Lines 307-310: One sentence that has been made into its own paragraph. Should incorporate this sentence with the following paragraph.
- Chose HMW wax lipids based on chain length (lines 376-379)
- Need to include Fig. 3B for reference in the parentheses in lines 376-379
- Typo: line 388, should say terrigenous d13C endmember, not marine
- They don't mention what lignin phenols they used
- Every time they reference figure 5 in the paper, they should be referring to figure 4 (example: 5A should be 4A)
- The authors then need to include Fig. 5 in the text of their paper once they made the changes to Fig. 4
- Figures 2-5 are also very descriptive. Leave the interpretation of the data to the discussion
- Their S/V and C/V explanation (section 3.2.2) should be taken with a grain of salt, the loss of C can make it look like woody material when it is not
- Acid/aldehyde values for the syringyl phenols off the shelf seem too high (Fig. 5A)
- Lines 513-516: statement does not seem accurate and it also applies to a different shelf system
- Lines 537-538: the chain length should not determine which lipids are HMW

---

## Author Comment (AC1) · 1 Aug 2016

**Author responses to reviews and edits to Biogeosciences manuscript bg-2016-159 "Fate of terrigenous organic matter across the Laptev Sea from the mouth of the Lena River to the deep sea of the Arctic interior"**

**by Lisa Bröder, Tesi, Salvadó, Semiletov, Dudarev and Gustafsson**

We are grateful to the two reviewers for their detailed and insightful comments on our manuscript.  We are naturally delighted of the overall supportive assessments. Their constructive reviews and suggestions have contributed to substantially improve the paper during our revisions. All referee comments and our responses, as well as the resulting edits, are detailed below, organized such that first the reviewer comments are given in italic, directly followed by our response and outline of the resulting edit in regular font. References in our response to line numbers refer to the revised manuscript version (with tracked changes).

**Reviewer #1**

GENERAL COMMENTS:

*"Degradation of terrestrial organic matter (TerrOC) along its transport into the ocean is an extensively investigated yet not fully understood aspect of the global carbon cycle. The Siberian Arctic Shelf, with a width > 800 km, is a unique and ideal place to study the transformation of TerrOC upon its entry into the sea. In particular, with climate-induced mobilization of permafrost-locked TerrOC, this area is receiving greater attention than many other shelves on earth. This paper uses a series of terrestrial biomarkers (including lignin phenols, cutin acids, and wax lipids) to study the abundances as well as degradation of TerrOC along a 800 km transect from Lena River mouth across the shelf to the slope and rise. In conjunction with carbon isotope and surface area analyses, the authors display an increasing TerrOC degradation with increasing distance from the coast. The dataset is large and unique, and the writing is clear and organized. I have a few suggestions for the authors to consider.*

*First, in the Results and Discussion, many comparisons are made to other published Arctic (or other non-Arctic) studies, which is great and necessary. But the text is a bit reiterative and I wonder if there is a better way to display all the information from literature with tables or figures more vividly, which will help the readers to digest.*
*Second, TerrOC degradation is not unique to permafrost-derived OC: it happens in other shelf environments without permafrost OC input. What I am interested in this study is what is special about the transformation of TerrOC in the Siberian Arctic Shelf in comparison with other parts of the world or other depositional environment. I think the authors have made some very nice comparisons with the Mackenzie Shelf. But I think this may be more emphasized in the conclusions, etc."*

RESPONSE:
We are glad about the positive appraisal and appreciate the clear, concise and constructive comments. We agree with the suggestion of a more illustrative way for the comparisons made with other studies and have now sought to improve on this. Therefore, we have

inserted an additional figure (Fig. 4), where the lignin phenol data of this study are compared to the literature data for several Arctic Shelf seas. For the other proxies (biomarker concentrations and ratios), we added similar figures in the Supplementary Information (Fig. S1 and S2). Here we recalculated the HMW lipid data to match the definition in previous studies (carbon chain lengths of ≥ 20 for *n*-alkanoic acids and ≥ 21 for *n*-alkanes) in order to be truly comparable (see also Response to Reviewer#2).

The Laptev Sea and adjacent East Siberian Sea are among the widest continental margins on Earth. The resulting long transport and time spent in oxic sediments exert first order control on the land-derived OC degradation. Our study area thus is a perfect natural laboratory to test hypotheses on the fate of permafrost carbon in terms of carbon-climate feedback. This is a key aspect which hadn't been sufficiently discussed in the Introduction and in the Conclusions. We have now updated the text accordingly (L75-79 and 634-637).

SPECIFIC POINTS:

1) *"L19: Change to "Mobilized permafrost carbon" can be either…"*

This has been changed accordingly in the text.

2) *"L147: How does combustion affect surface area measurement?"*

From the original text it was not clear that the combustion was done to remove organic material in order to obtain the mineral specific surface area. According to Keil and Cowie (1999), this particular method does not alter the surface area systematically compared to the removal of organic matter with sodium pyrophosphate/ hydrogen peroxide as in Mayer (1994). This explanation has been inserted in the text (L154-157).

3) *"L165: What about FeOx, which plays a key role in the preservation of TerrOC?"*

First of all, the iron quantified here refers to the bulk iron which includes both fractions associated with OM and detrital material (e.g. clay material). In addition, XRF results are reported only as $Fe_2O_3$ as samples are combusted (450C, 12h) prior to the analysis to remove the organic fraction. Thus, we decided not to include the iron data obtained from the XRF measurements here, as we cannot determine the fraction of amorphous iron oxide/hydroxides (of high surface area) with this method and therefore do not obtain any information of the association between Fe and organic matter. Even though not in the focus of this study, we do acknowledge the importance of FeOx in the Laptev Sea and have therefore added a reference to the study by Salvadó et al. (2015), where the association between iron and organic matter on the ESAS is studied in more detail (L67-68).

4) *"L346-348: How does hydrodynamic sorting affect the SA-normalized abundance of lignin? This is probably an important aspect (if not more important) other than degradation, which may explain the varied decrease rate for lignin vs. wax lipids. Does this bias the SA-normalized abundances?"*

This is indeed an important aspect that we have tried to address with the help of the study by Tesi et al. (2016), which had the objective of disentangling exactly these two processes: degradation and hydrodynamic sorting during cross-shelf transport. We have

now elaborated more on the matter and included some rough estimates of how much of the lignin decrease can be attributed to degradation vs sorting (L449-459). According to Tesi et al. (2016) most of the sorting is occurring in close proximity to the coast, i.e. water depths of less than 25 m. Assuming that for the shallowest station up to 75 % of the lignin phenols are associated to the low density/large plant fragment fraction and thus retained close to the shore, we can "correct" for sorting by focusing on the lignin phenols in the fine, mineral associated fraction (reducing the original value for the bulk to 25 %, i.e. 11 instead of 45 µg m$^{-2}$). The cross-shelf loss from about 11 µg m$^{-2}$ to 0.43 ± 0.09 µg m$^{-2}$ still corresponds to a reduction by 96 ± 1 % that can be attributed to degradation (instead of 99.1 % net loss), which agrees with the values presented in Tesi et al. (2016).

**Reviewer #2**

GENERAL COMMENTS:

*"• Does fill a much needed role as it is one of few papers that looks at the fate of terrigenous organic matter as it is carried out past the continental shelf*

*• They do compare their data to other shelf studies but some of there comparisons are not valid (i.e. explaining differences in HMW degradation in different studies is due to differences in chain length)*

*• Seems likely that the amount of time spent during cross-shelf transport is correlated with terrigenous organic matter degradation"*

RESPONSE:
We appreciate and are encouraged by the positive assessment, yet have paid attention to the constructive criticism and suggestions. In particular, we have recalculated the results for the HMW *n*-alkanes and *n*-alkanoic acids using the same chain-length definition as in the studies we are comparing our values to. These values are then compared in the new figures S1 and S2.

SPECIFIC POINTS:

1) *"Figure 1 should include coastal currents and could have an inset of where the Laptev Sea is relative to the rest of the arctic"*

Figure 1 has been changed accordingly.

2) *"Need to justify in the paper that the terrigenous matter in this study is only coming from the Lena River and not from the two other rivers (shown in Figure 1 and are not labeled) that empty into the Laptev Sea"*

The other two rivers in Fig. 1, Olenjok and Anabar (names have now been added to the map), discharge combined only less than 7 % of water and total suspended matter (TSM) of the total for the Laptev Sea (less than 10 % of that of the Lena River) according to Gordeev (2004). The second largest river discharging to the Laptev Sea, Khatanga (not in the map, to the west of Anabar), contributes <12 % of total water and <6 % of TSM (~16 % and ~8 % of that of the Lena River, respectively). We therefore assumed that the largest fraction of riverine-delivered TerrOC should originate from the Lena River (>70 %

of both water and TSM discharge to the Laptev Sea) for both water and TSM. The overwhelming importance of the Lena River as sediment source to the Laptev Sea has now been more emphasized in the text (L101-102).

3) *"Need specific references when discussing what TOC/SA ratios are expected for what kind of environment (i.e. river, deep ocean; lines 248-253)."*

The corresponding references (Mayer, 1994; Mayer et al., 2002; Aller and Blair, 2006) have been added to the text (L264, L267, L268).

4) *"Line 272: Uses a lateral transport time for an active margin instead of one of a passive margin. Suggest using a east coast system from the U.S."*

Unfortunately we could not find any quantitative constrains for lateral transport times of OC across other margins. The numbers presented by Keil et al. (2004) should serve only as a rough estimate of the timescale to test if the explanation of ageing during transport could hold. We state in the text that transport across the wide Laptev Sea is expected to take much longer.

5) *"Along the same lines as bullet point number 2 in this section, if you cannot prove the source of this OC is the same, then you can not prove that it is aging"*

As stated earlier, the main POC sources for the Laptev Sea are coastal erosion and Lena River, followed by a much smaller contribution from marine sources. Preferential degradation of the modern source (marine) could also shift the 14C signature towards lower (older) values. This alternative explanation, yet not supported by the d13C values, has been added to the text (L293-301). Another piece of evidence for protracted transport is the highly reworked condition of the terrestrial material as shown by all degradation proxies, which are discussed in Section 3.3. We also have unpublished stable hydrogen isotope ratio data on HMW *n*-alkanes (in other manuscript in advanced stage for submission), which do not show any significant difference between the shelf and slope/rise sediments. These results suggest that there is no significant change of input material or preferential degradation of coastal erosion (yedoma) vs riverine TerrOC between shelf and slope/rise.

6) *"Lines 307-310: One sentence that has been made into its own paragraph. Should incorporate this sentence with the following paragraph."*

This has been changed accordingly in the text.

7) *"Chose HMW wax lipids based on chain length (lines 376-379)"*

In this study, we are only reporting concentrations of HMW *n*-alkanes and *n*-alkanoic acids. For these compounds, due to their simple chemical structure, the carbon chain-length determines the molecular weight. The terms "long-chained" and "HMW" may therefore be used interchangeably. We have now moved this explanation and definition (cutoff for HMW *n*-alkanes 23 carbon atoms, for HMW *n*-alkanoic acids 24 carbon atoms) to the methods section (L234-236).

8) *"Need to include Fig. 3B for reference in the parentheses in lines 376-379"*

This has been inserted.

9) *"Typo: line 388, should say terrigenous d13C endmember, not marine"*

Thank you for pointing this out. It has been changed accordingly.

10) *"They don't mention what lignin phenols they used"*

We have now included this information (L369-373).

11) *"Every time they reference figure 5 in the paper, they should be referring to figure 4 (example: 5A should be 4A)"*

Since we have inserted a new figure 4, now these references are actually correct and have therefore not been changed.

12) *"The authors then need to include Fig. 5 in the text of their paper once they made the changes to Fig. 4"*

References to this figure (now Fig. 6) have been included in the text (L545 and 582).

13) *"Figures 2-5 are also very descriptive. Leave the interpretation of the data to the discussion"*

We assume that this comment is directed at the relatively long captions of Fig. 2-5. We have now shortened those substantially and removed any interpretations of the displayed data.

14) *"Their S/V and C/V explanation (section 3.2.2) should be taken with a grain of salt, the loss of C can make it look like woody material when it is not"*

We have explicitly mentioned this alternative explanation for the observed decrease in C/V ratios in the text (L498-500). For S/V ratios, however, we observe the opposite trend (increasing with increasing water depth) as found for degradation by fungi (decreasing, Hedges et al., 1988).

15) *"Acid/aldehyde values for the syringyl phenols off the shelf seem too high (Fig. 5A)"*

We have double-checked our results for the acid/aldehyde values and did not find anything unusual (similar response for all, peak shapes look fine, data are above quantification limit). Moreover, our findings seem to be consistent with other studies that reported values in a similar range (e.g. Winterfeld et al., 2015), see also new figure S1.

16) *"Lines 513-516: statement does not seem accurate and it also applies to a different shelf system"*

One of the three transects studied in Tesi et al. (2016) (called W, located in the Laptev Sea) is actually part of the (longer) transect investigated here. But we had apparently over-simplified their findings for the acid/aldehyde ratios. This has been rectified (L556-562).

17) *"Lines 537-538: the chain length should not determine which lipids are HMW"*

As stated in the response to point 7, in the case of *n*-alkanes and *n*-alkanoic acids the molecular weight directly depends on the chain length. We have therefore used the chain lengths for the definition of HMW.

We have also discovered during the review process, that the water depths of two stations had been listed slightly wrong (TB-46 and SW-6: 6 instead of 11 m and 92 instead of 89 m, respectively) and have updated all figures and tables accordingly. All other changes are minor.

---

## Author Response (AR2)

**Author responses to minor revisions to Biogeosciences manuscript bg-2016-159 "Fate of terrigenous organic matter across the Laptev Sea from the mouth of the Lena River to the deep sea of the Arctic interior"**

**by Lisa Bröder, Tesi, Salvadó, Semiletov, Dudarev and Gustafsson**

We are glad to hear the editor's positive assessment of our revised manuscript, yet have taken the constructive criticism seriously. All comments and our responses, as well as the resulting edits, are detailed below, organized as in the previous response letter: first the reviewer comments are given in italic, directly followed by our response and outline of the resulting edit in regular font. References in our response to line numbers refer to the revised manuscript version (with tracked changes).

**Editor comments**

SPECIFIC POINTS:

1) *"L76-78: The order of words seems a bit off to me. Please check the phrasing."*

The original *"The resulting long cross-shelf transport and thereby time spent in oxic sediments might exert first order control on TerrOC degradation…"* has been reworded to *"Protracted cross-shelf transport may hence result in long oxygen exposure times, which might exert first order control on TerrOC degradation…"*

2) *"L301: input of young autochtonous material is not supported by the stable carbon isotopic signature – please specify how."*

The explanation has now been added in L302-304: *"However, this latter scenario is not supported by the stable carbon isotopic signature, as values for δ13C increase from about -24.3 ‰ on the mid-shelf to about -22.5 ‰, suggesting a higher fraction of marine organic matter for the deep stations."*

3) *"Comment 4 of reviewer 2: I still think this is a valuable comment. Please specify that the transport times reported by Keil et al. (2004) are for an active margin, and should be taken here as a minimum."*

This has been clarified in L290-292.

4) *"Add the contribution of the reviewers to the acknowledgements."*

The work of the reviewers has now been acknowledged in L659-661.

5) *"Table 1: have d13C measurements done in duplicate? If so, what was the error, and what value is displayed here in the table (average)? What is the precision of the measurement, i.e. can you report d13C values with this amount of significant numbers?"*

The d13C values stem from single measurements. Since the reproducibility of the instrument is <0.15 ‰, there should not be more than three significant digits in the table. This has now been adjusted accordingly.

[revised manuscript text omitted]